

# Genome-wide identification and expression analysis of the *14-3-3* gene family in soybean (*Glycine max*)

Yongbin Wang[1,2], Lei Ling[3], Zhenfeng Jiang[2], Weiwei Tan[1], Zhaojun Liu[1], Licheng Wu[1], Yuanling Zhao[4], Shanyong Xia[4], Jun Ma[4], Guangjin Wang[5] and Wenbin Li[2]

[1] Biotechnology Research Institute, Heilongjiang Academy of Agricultural Sciences, Harbin, Heilongjiang, China
[2] Key Laboratory of Soybean Biology in Chinese Ministry of Education, Key Laboratory of Soybean Biology and Breeding/Genetics of Chnese Agriculture Ministry, Northeast Agricultural University, Harbin, Heilongjiang, China
[3] Harbin Normal University, Harbin, Heilongjiang, China
[4] Heilongjiang Academy of Agricultural Sciences, Harbin, Heilongjiang, China
[5] Soybean Research Institute, Heilongjiang Academy of Agricultural Sciences, Harbin, Heilongjiang, China

## ABSTRACT

In eukaryotes, proteins encoded by the *14-3-3* genes are ubiquitously involved in the plant growth and development. The *14-3-3* gene family has been identified in several plants. In the present study, we identified 22 *GmGF14* genes in the soybean genomic data. On the basis of the evolutionary analysis, they were clustered into $\varepsilon$ and non-$\varepsilon$ groups. The *GmGF14s* of two groups were highly conserved in motifs and gene structures. RNA-seq analysis suggested that *GmGF14* genes were the major regulator of soybean morphogenesis. Moreover, the expression level of most *GmGF14s* changed obviously in multiple stress responses (drought, salt and cold), suggesting that they have the abilities of responding to multiple stresses. Taken together, this study shows that soybean *14-3-3s* participate in plant growth and can response to various environmental stresses. These results provide important information for further understanding of the functions of *14-3-3* genes in soybean.

## INTRODUCTION

The *14-3-3* genes are first isolated from brain tissue, and they are ubiquitously found in eukaryotes (*Li et al., 2015*; *Yang et al., 2017*; *Kumar et al., 2015*; *Takahashi, 2006*). *14-3-3* proteins are highly conserved small acidic proteins in different organisms, encoded by a large gene family (27–32 kDa) (*Ferl, Lu & Bowen, 1994*; *Cao & Tan, 2018*). These proteins can form dimers (homo- or hetero- dimers) and have approximately nine antiparallel $\alpha$-helices (*Ferl, Manak & Reyes, 2002*; *Rodriguez & Guan, 2010*). These structures work as binding sites and interact with *14-3-3* proteins and their target, and they also bring two proteins together as a protein complex based on their dimeric properties (*Sijbesma et al., 2017*; *Valente et al., 2012*; *Li & Dhaubhadel, 2012*). 14-3-3s are involved in several protein-protein interactions, such as responding to biotic/abiotic stress, participating in

Corresponding authors
Guangjin Wang, gjw1962@yeah.net, wenbinli@yahoo.com
Wenbin Li, wenbinli@neau.edu.cn

plant hormone signaling and regulating tissue development in various plants (*Roberts, Salinas & Collinge, 2002*; *Camoni et al., 2018*; *Zhang et al., 2010*).

To date, more *14-3-3s* have been reported in several plants, such as Arabidopsis, rice, tobacco, populus and *Medicago truncatula* (*Chen et al., 2006*; *Rosenquist et al., 2001*; *Xu & Shi, 2006*; *Tian et al., 2015*; *Cheng et al., 2016*). In plants, the *14-3-3* proteins were named as GF14 or GRF due to they are a part of protein/G-box complex (*De Vetten & Ferl, 1994*; *Rosenquist et al., 2001*). *14-3-3s* are distributed in different organelles, such as cytoplasm, cell membrane, nucleus, chloroplast and mitochondria (*Bihn et al., 2010*; *Sehnke et al., 2000*; *Ferl, Manak & Reyes, 2002*). In plants, they were regulated by several biological processes (*Cheng et al., 2016*; *Tian et al., 2015*); for example, multiple mutant analysis suggested that Arabidopsis *14-3-3* genes regulate root growth, chloroplast division, photosynthesis and leaf longevity (*Liesbeth et al., 2015*). *GhGRFs* were found involving in plant development and signaling transduction in cotton fiber (*Zhang et al., 2010*). In addition, an increasing number of works were carried out to investigate the roles of *14-3-3s* in plants under multiple stresses (*Roberts, Salinas & Collinge, 2002*). Most of the *OsGRF* genes' expression changed under heat, cold and salt stresses (*Yashvardhini et al., 2017*). The results of overexpression of *AtGRF6* in transgenic cotton showed a stay-green phenotype, indicating that they can improve plant tolerance to drought stress (*Juqiang et al., 2004*).

Soybean is an important cash crop in the world, while its production is often influenced by various environmental stresses (*Masuda & Goldsmith, 2009*). However, not enough attention is focused on soybean *GmGF14s*. Previous studies identified 18 *GmGF14* genes in soybean, which was different from our study (*Li & Dhaubhadel, 2011*). In this study, we identified a total of 22 *GmGF14* genes in soybean genome. Phylogenic relationship, gene structures, protein motifs and expression patterns of all the *GmGF14* genes were analyzed, together with their responses to various stresses in soybean. These results will provide important information for further study to understand the regulating mechanism of *GmGF14* genes during their growth and their responding abilities to various environmental stresses.

## MATERIALS & METHODS

### Identification of *14-3-3* genes

The Hidden Markov Model (HMM) profiles of the *14-3-3* motif (PF00244) were downloaded from the Pfam database (*Punta et al., 2004*). HMM searched *14-3-3* motif (PF00244) from the *Glycine max* protein database with values (*e*-value) cut-off at 0.1 (*Punta et al., 2004*). The integrity of the *14-3-3* motif was determined using the online program SMART (http://smart.embl-heidelberg.de/) with an e-value < 0.1 (*Ivica, Tobias & Peer, 2012*). In addition, the three fields (length, molecular weight, and isoelectric point) of each *14-3-3* protein was predicted by the online ExPASy program (http://www.expasy.org/tools/) (*Johana et al., 2015*).

### Phylogenetic analysis

To investigate the phylogenetic relationship of the *14-3-3* gene families in *Arabidopsis thaliana*, *Oryza sativa*, *Medicago truncatula* and *Glycine max*, *14-3-3* protein sequences were

downloaded from Phytozome v12.1 (http://www.phytozome.org) (*Goodstein et al., 2012*). 14-3-3s were aligned using the BioEdit program. A neighbor-joining (NJ) phylogenetic tree was constructed using the MEGA 5.0 program (*Tamura et al., 2011*). Bootstrapping was performed with 1000 replications. Genes were classified according to the distance homology with *Arabidopsis thaliana* genes (*Ferl, Lu & Bowen, 1994*).

## Sequence alignment, motif prediction and gene structure of *14-3-3* genes

The 3D structure of *14-3-3* proteins were predicted by using Phyre[2] and ESPript 3.0 software (*Gouet, Robert & Courcelle, 2003*; *Kelley et al., 2015*). Multiple alignments of proteins were conducted using Jalview software with ClustalW method (*Clamp et al., 2004*). The online MEME analysis is used to identify the unknown conserved motifs (http://meme.ebi.edu.au/) by using the following parameters: site distribution: zero or one occurrence (of a contributing motif site) per sequence, maximum number of motifs: 20, and optimum motif width $\geq$6 and $\leq$200 (*Bailey et al., 2015*). A gene structure display server program (http://gsds.cbi.pku.edu.cn/) was used to display the *G. max 14-3-3* gene structures.

## Gene duplication and collinearity analysis

The physical locations of the *GmGF14* genes on the soybean chromosomes were mapped by using MG2C website (http://mg2c.iask.in/mg2c_v2.0/). The analysis of synteny among the soybean genomes was conducted locally using a method similar to that developed for the PGDD (http://chibba.agtec.uga.edu/duplication/) (*Krzywinski & Schein, 2009*). First, BLASTP, OrthoMCL software (http://orthomcl.org/orthomcl/about.do#release) and MCScanX software (*Wang et al., 2012*) were used to search for potential homologous gene pairs (E $<1$ $e^{-5}$, top 5 matches) across multiple genomes. Then, these homologous pairs were used as the input for the PGDD database (http://chibba.agtec.uga.edu/duplication/). Ideograms were created by using Circos (*Krzywinski & Schein, 2009*).

## Calculating *Ka* and *Ks*

The *Ka* and *Ks* were used to assess selection history and divergence time (*Li, Gojobori & Nei, 1981*). The number of synonymous (*Ks*) and nonsynonymous (*Ka*) substitutions of duplicated *14-3-3* genes were computed by using the KaKs_Calculator 2.0 with the NG method (*Wang et al., 2010*). The divergence time (*T*) was calculated using the formula $T = Ks/ (2\times 6.1\times 10^{-9}) \times 10^{-6}$ million years ago (MYA) (*Kim et al., 2013*).

## *14-3-3* genes expression analysis of soybean

The expression data of *14-3-3* genes in different tissues, including root, root hair, flower, nodule, pod, stem, leaf, SAM and seed, were available in Phytozome v12.1 database (https://phytozome.jgi.doe.gov/pz/portal.html). We retrieved the fragments per kilobase per million reads (FPKM) value representing the expression level of *GmGF14* genes, then generated the heatmap and k-means clustering by using R 3.2.2 software.

## Plant material and treatments

*Glycine max* (Williams 82) was used in this study. Seeds were planted in a 3:1 (w/w) mixture of soil and sand, germinated, and irrigated with half-strength Hoagland solution once every 2 days. The seedlings were grown in a night temperature of 20 °C and day temperature of 22 °C, relative humidity of 60%, and a 16/8 h photoperiod (daytime: 05:00–21:00). After 4 weeks, the germinated seedlings were treated with 20% PEG6000 (drought), 250 mM NaCl solution (salt), and 4 °C (cold). Control and treated seedlings were harvested 1 h, 6 h, and 12 h after treatment. All samples were frozen in liquid nitrogen and stored at −80 °C until use.

## RNA extraction and Quantitative real-time PCR (qRT-PCR)

Total RNA was extracted from the root of soybean using RNAiso Plus (TaKaRa, Toyoto, Japan) according to manufacturer's instructions. 2 μg RNA was extracted using PrimeScript RT reagent Kit with gDNA Eraser (TaKaRa,Toyoto, Japan). The cDNA samples were diluted to 2.5 ng/L. Quantitative Real-time PCR (qRT-PCR) was performed using SYBR *Premix Ex Taq* II (TaKaRa, Toyoto, Japan) on an ABI Prism 7000 sequence detection system (Applied Biosystems, USA) with the primers listed in Table S1. PCR amplification was performed in accordance with SYBR *Premix Ex Taq* (TaKaRa, Toyoto, Japan) response system. For each sample, three biological replicates were conducted. Relative expression was calculated by the $2^{-\Delta\Delta Ct}$ method (*Livak & Schmittgen, 2001*). The *actin* and *GAPDH* genes were used as internal control.

## Gene ontology enrichment

Once the sequences were obtained ran a BLASTX search against the Uniref100 database at a 1e−30 significance level. The matches were extracted and compared to the GO annotation generated against Uniref100 hits located at EBI. The GO annotation of the *GmGF14* genes by using WEGO 2.0 website (http://wego.genomics.org.cn/).

## RESULTS

### Identification and multiple sequences alignment of *GmGF14* genes

We identified 22 GmGF14 genes, which were named from GmGF14a to GmGF14v based on their physical locations on chromosomes. ExPASy predicted that 22 GmGF14 proteins have different physical and chemical properties as their amino acid lengths ranged from 71 aa (GmGF14v) to 754 aa (GmGF14j), with an average of 295 aa, and their molecular weights ranged from 7.92 kDa (GmGF14v) to 81.75 kDa (GmGF14j), the isoelectric points ranged from 4.67 (GmGF14c/e) to 5.7 (GmGF14v). Detail information of GmGF14 proteins is provided in Table 1. Besides, we found that most of the GmGF14s contain highly conserved domains, and ten α-helices were identified in their secondary structures (Fig. 1). In addition, the C-terminal end of *14-3-3* proteins are quite unique in sequence and length.

### Phylogenetic analysis of the GmGF14s

We constructed a phylogenetic tree to show the phylogenetic and evolutionary relationships of *GmGF14* genes among *A. thaliana*, *O. sativa*, *M. truncatula* and *G. max* (Fig. 2). The

**Table 1  List of all *GmGF14* genes identified in the *Glycine max* genome.**

| Gene name | Gene locus | Chromosome location | Length (aa) | pI | Molecular weight (Da) | Group |
|---|---|---|---|---|---|---|
| GmGF14a | Glyma.01G058000 | Chr01:7642485-7646277 | 260 | 4.83 | 29,487.17 | ε |
| GmGF14b | Glyma.02G115900 | Chr02:11280858-11285984 | 260 | 4.83 | 29,498.19 | ε |
| GmGF14c | Glyma.02G208700 | Chr02:39388574-39391014 | 263 | 4.67 | 29,353.86 | non-ε |
| GmGF14d | Glyma.04G092600 | Chr04:8158031-8160711 | 251 | 4.81 | 28,208.84 | non-ε |
| GmGF14e | Glyma.04G099900 | Chr04:9132954-9135203 | 289 | 4.67 | 32,432.54 | non-ε |
| GmGF14f | Glyma.04G183400 | Chr04:45129363-45133820 | 727 | 4.94 | 79,220.24 | ε |
| GmGF14g | Glyma.05G158100 | Chr05:35025422-35029392 | 260 | 4.8 | 29,249.8 | ε |
| GmGF14h | Glyma.06G094400 | Chr06:7432085-7434388 | 251 | 4.81 | 28,365.07 | non-ε |
| GmGF14i | Glyma.06G101500 | Chr06:8052625-8054939 | 280 | 5.46 | 31,708.21 | non-ε |
| GmGF14j | Glyma.06G182800 | Chr06:15705290-15709591 | 754 | 5.06 | 81,749.77 | ε |
| GmGF14k | Glyma.07G226000 | Chr07:40298318-40302692 | 260 | 4.79 | 29,579.23 | ε |
| GmGF14l | Glyma.08G115800 | Chr08:8877809-8881104 | 260 | 4.9 | 29,247.74 | ε |
| GmGF14m | Glyma.08G363800 | Chr08:47528826-47532060 | 261 | 4.81 | 29,384.49 | non-ε |
| GmGF14n | Glyma.12G210400 | Chr12:36943077-36946491 | 262 | 4.73 | 29,461.02 | ε |
| GmGF14o | Glyma.12G229200 | Chr12:38919217-38923409 | 266 | 4.85 | 30,493.26 | ε |
| GmGF14p | Glyma.13G270600 | Chr13:37265741-37269626 | 264 | 4.84 | 30,207.93 | ε |
| GmGF14q | Glyma.13G290900 | Chr13:39120795-39124124 | 262 | 4.77 | 29,518.07 | ε |
| GmGF14r | Glyma.14G176900 | Chr14:43637893-43642553 | 315 | 4.71 | 35,233.85 | non-ε |
| GmGF14s | Glyma.17G208100 | Chr17:34108328-34108849 | 160 | 5.61 | 18,687.61 | non-ε |
| GmGF14t | Glyma.18G298300 | Chr18:57587135-57590454 | 258 | 4.7 | 29,063.69 | non-ε |
| GmGF14u | Glyma.20G025900 | Chr20:2845106-2852380 | 261 | 4.79 | 29,640.22 | ε |
| GmGF14v | Glyma.20G043700 | Chr20:7939112-7939943 | 71 | 5.7 | 7920.14 | ε |

22 GmGF14 proteins were composed of ε group or non-ε group, 13 GmGF14 proteins (GmGF14a/b/f/g/j/k/l/n/o/p/q/u/v) belonged to the former group, while the other 9 GmGF14 proteins (GmGF14c/d/e/h/i/m/r/s/t) belonged to the non-ε group.

## Gene structure and motif analysis

Exon/intron pattern divergence plays a crucial role during evolution. We analyzed the exon/intron pattern of *GmGF14s* and found that genes of soybean contained 1–6 introns. Among them, non-ε group *GmGF14* genes contained 1–4 introns, whereas ε group genes had 1–6 introns. The exon/intron pattern were obviously different in the two groups of *GmGF14* genes, suggesting the diversity of *GmGF14* genes during the evolution (Fig. 3A, Tables S2 and S3). A total of 15 conserved motifs in *GmGF14* genes were identified by MEME software. As shown in Fig. 3B, 5 motifs (motifs 1–5) were annotated as *14-3-3* domains, and most of GmGF14 proteins contained these motifs. All non-ε group GmGF14 proteins shared motifs 3, 4 and 15, whereas most ε group soybean 14-3-3 proteins contained the motifs 1–7 and motif 15. In addition, the GmGF14f/j in ε group contained motifs 8-14, and GmGF14v only had motif 6.

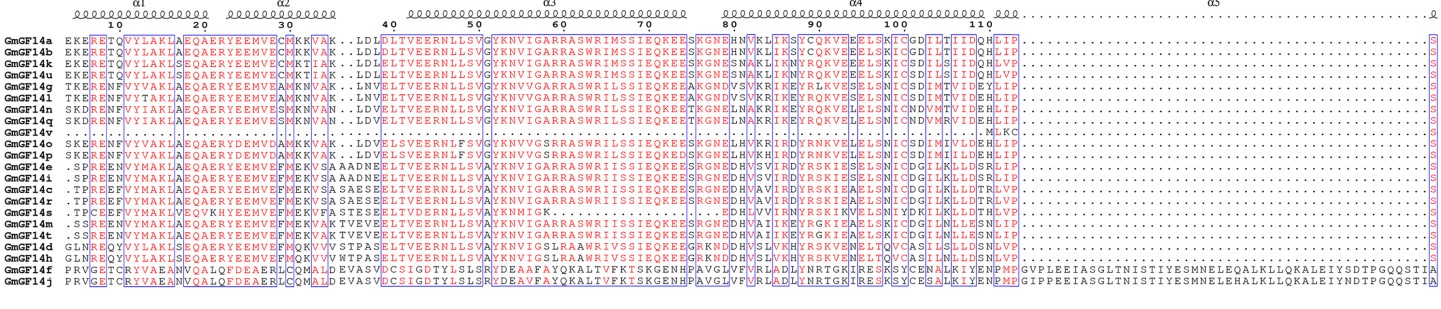

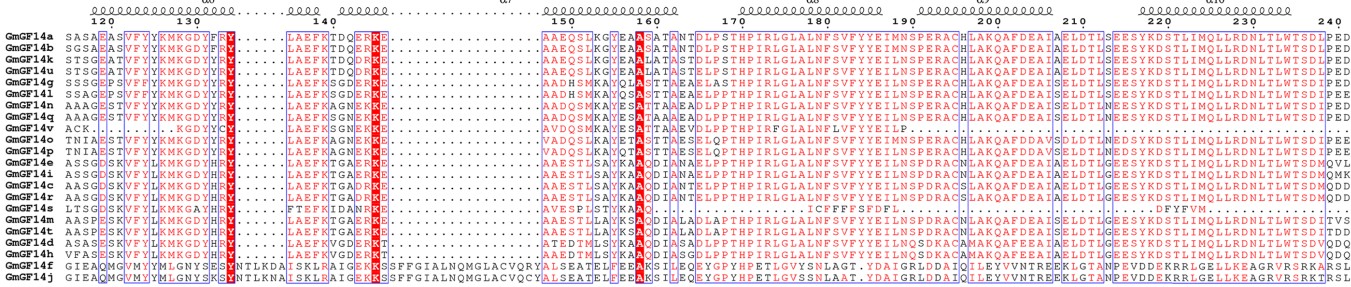

**Figure 1** Multiple sequence alignment of all 14-3-3 proteins from soybean.

## Chromosomal location and duplication analysis

A chromosomal location map of *GmGF14* genes on each chromosome was drawn. As shown in Fig. 4, 22 *GmGF14* genes were mapped to thirteen of twenty chromosomes unevenly, and they were densely distributed on chromosome 4 and chromosome 6, containing 3 members, respectively (Fig. 4). Most of them were distributed on the two ends of the chromosomes. To better understand the evolution of soybean *14-3-3* genes, we checked genome duplication events in this gene family. The *GmGF14* gene pairs had 19 segmental duplication events without tandem duplication (Table S4). Among the duplication events, genes on chromosome 8 had the largest number of that (Fig. 4).

## Evolution and divergence of the *14-3-3* gene family

We found 19 pairs of paralogous in soybean, 27 orthologous pairs between soybean and Arabidopsis, 15 orthologous pairs in soybean and *M. truncatula* (Table 2). Additionally, we found that two *14-3-3* genes (*GmGF14i* and *GmGF14s*) did not have any homology genes. Two or more *GmGF14* genes matched to one *AtGRF* gene or *Mt14-3-3* gene, implying that these genes might play key roles in the *GmGF14* genes' expansion during evolution. In addition, to examine the evolutionary selection process, we calculated *Ka/Ks* ratios of 19 *GmGF14* paralogous pairs (Table 3). All the *Ka/Ks* value were under 0.3, indicating that they had evolved mainly in strong purifying selection. The gene differentiation of the 19 gene pairs were approximately occurred in the 5–20 MYA.

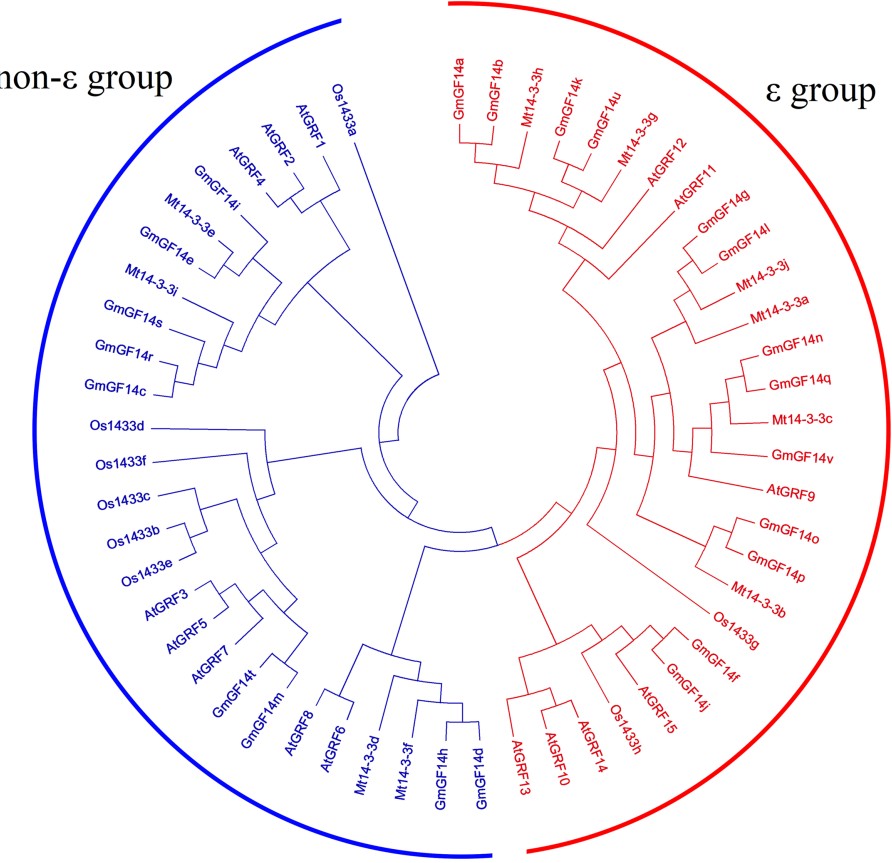

**Figure 2 Phylogenetic tree analysis of the *14-3-3* genes in *Glycine max, Arabidopsis thaliana, Medicago truncatula* and *Oryza sativa*.** The phylogenetic tree was constructed using MEGA 5.0 by the neighbor-joining method. The Bootstrap value was 1,000 replicates. The two clusters were designated as non-ε group and ε group, and indicated them in a specific color.

## *Cis*-elements in *GmGF14s* promoters

*Cis*-elements involved in transcriptional regulation and can response to variety stresses. We isolated the sequence which is 1.5 kb upstream of the *GmGF14* genes to explore their potential function (Table 4). We found nine potential elements, such as ABRE, AuxRR-core, GARE-motif, CGTCA/TGACG-motif, P-box, TATC-box, TCA-element and TGA-element, were involved in ABA (abscisic acid), IAA (auxin), GA (gibberellin), MeJA (methyl jasmonate) and SA (salicylic acid) regulating mechanism. Additionally, there were four elements (TC-rich repeats, ARE, MBS and LTR) involved in defense/stress, anaerobic induction, drought and low-temperature responses, respectively. In the *GmGF14* promoters, we found different types and numbers of *cis*-elements, indicating that they participated in different regulatory mechanisms during plant growth and development.

## Expression analysis of *GmGF14* genes in different tissues

We analyzed the expression level of *GmGF14* genes in different soybean tissues and organs (e.g., root, root hair, flower, nodule, pod, stem, leaf, SAM and seed) based on RNA-seq

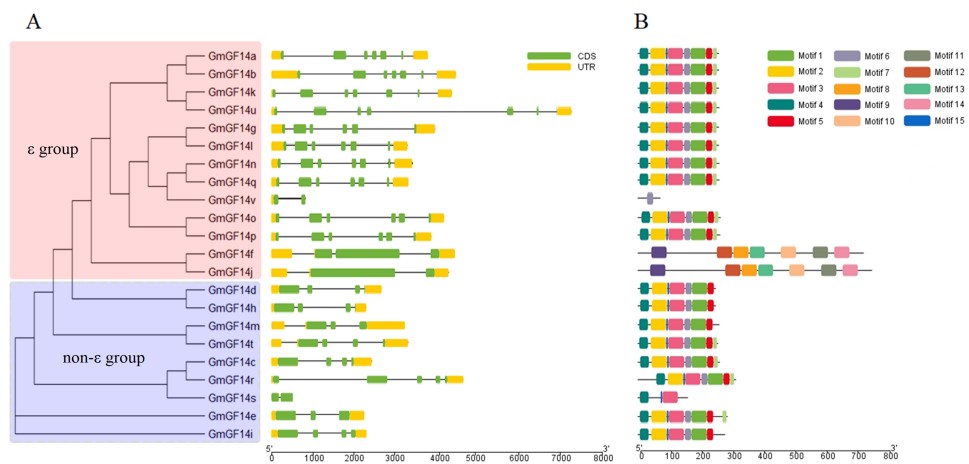

**Figure 3** **Conserved motifs and gene structure in *GmGF14s*.** (A) Exon/intron structures of *GmGF14* genes. (B) Conserved motifs of the GmGF14s. Each motif is represented by a number in colored box.

data (Fig. 5). Results showed that the expression level of most *GmGF14* genes varies in different tissues, suggesting the diversity of their roles. Significantly, most *GmGF14s'* expression level in vegetative organs (e.g., root, root hair, stem, leaf, and SAM) were higher than that of reproductive organs (e.g., flower, pod and seed). Ten *GmGF14* genes (*GmGF14e/i/h/c/r/m/n/q/g/t*) were highly expressed in all tested tissues, suggesting that they regulated the growth and development of soybean. *GmGF14k* was specifically expressed in root, *GmGF14p* was highly expressed in pod and stem, *GmGF14o* was highly expressed in pod. In addition, *GmGF14s* and *GmGF14v* can't be detected in these tissues.

## Expression patterns of *GmGF14s* under abiotic stress

Drought, salinity and cold are major factors affecting the production of soybean under natural conditions. We selected 19 genes from *GmGF14* which were differentially expressed (except *GmGF14s, GmGF14u* and *GmGF14v*) to further explore their expression pattern by qRT-PCR under abiotic stresses (Figs. 6–8, Tables S5–S8). The expression levels of them were changed over time during the stresses, showing that there were dynamic processes in the *GmGF14s* when they responding to stresses. In addition, we found that different gene duplication pairs have different expression patterns under the abiotic stresses (Table S5). During drought treatment, the expression pattern of three genes (*GmGF14a/b/g*) were all up-regulated in all the time point (Fig. 6, Table S6). Five genes (*GmGF14a/b/e/g/t*) were highly induced at 1 h, while the expression levels of *GmGF14c/d/q/r* did not significantly changed at all the time after drought treatment. Conversely, under drought treatment, four *GmGF14* genes (*GmGF14c/d/q/r*) were obviously down-regulated. Under salt stress, the expression levels of *GmGF14a/b/c/e/g/i/q/r/t* at 1 h time point up-regulated considerably, then the expression levels decreased (Fig. 7, Table S7). The expression levels of three *GmGF14* genes (*GmGF14f/h/p*) were generally down-regulated at one time point. 5 of 19 *GmGF14* genes (*GmGF14g/h/l/m/t*) were expressed essentially identical, with expression peaking at the first time point (1h) under cold stress (Fig. 8, Table S8). Eight genes

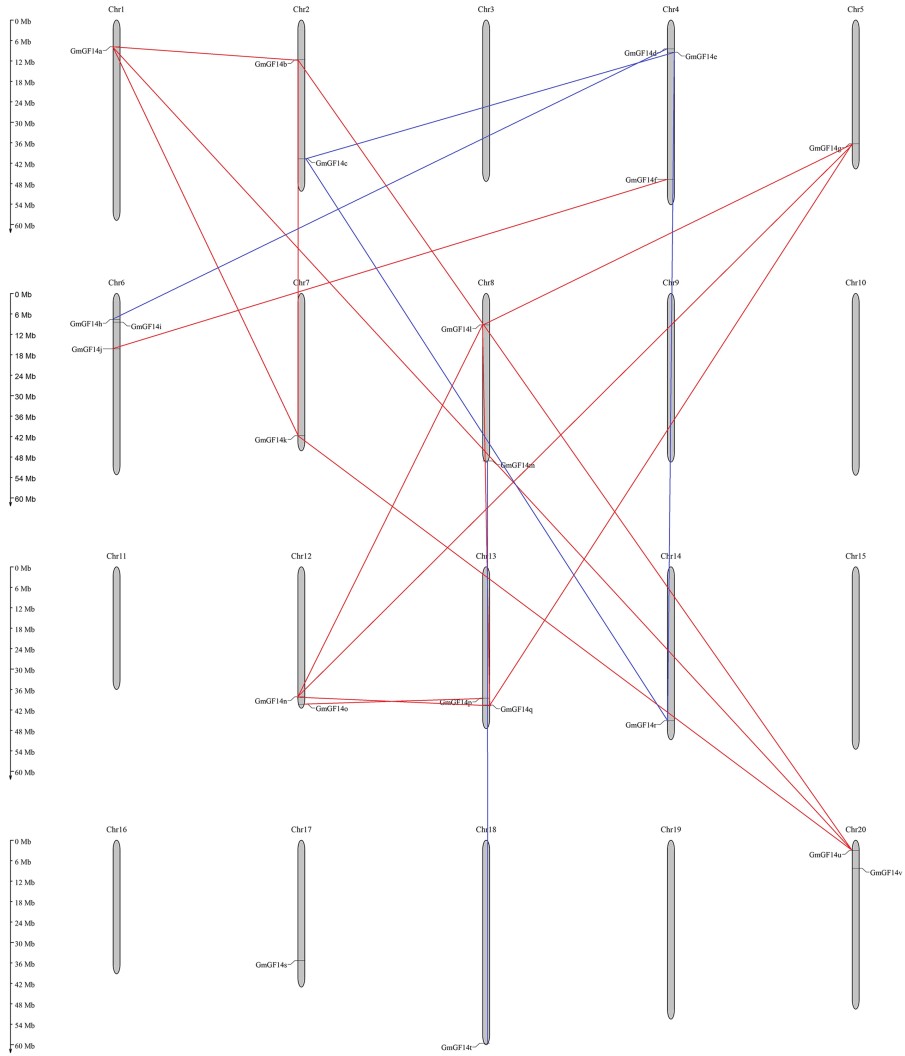

**Figure 4  Chromosome location and duplication events analysis in *Glycine max*.**

(*GmGF14a/b/c/d/e/n/q/r*) were not significantly changed at first time points (1h), followed by a strongly decrease under cold stress.

## Gene ontology enrichment

To further understand the functions of the GmGF14s, we performed GO annotation and GO enrichment analyses (Fig. S2 and Table S9). Usually, the GO terms included biological process, molecular function and cellular component. Within the biological process, most genes were assigned to the cellular process (16/22), single-organism process (16/22), biological regulation (14/22), regulation of biological process (14/22) and response to stimulus (14/22). In terms of cellular process, most GmGF14 proteins were predicted to be involved in cell (20/22), cell part (20/22), membrane (16/22), organelle (16/22) and organelle part (14/22). Few genes had predicted distributions in cell junction (2) and symplast (2). Within the molecular function category, only 20 GmGF14 proteins were

**Table 2  List of paralogous and orthologous pairs between soybean and *Arabidopsis thaliana* and *Medicago truncatula*.** Paralogous (Gm-Gm) and orthologous (Gm-Mt and Gm-At) gene pairs.

| Gm-Gm | Gm-Mt | Gm-At |
|---|---|---|
| GmGF14a/GmGF14b | GmGF14c/Mt14-3-3i | GmGF14f/AtGRF16 |
| GmGF14a/GmGF14k | GmGF14r/Mt14-3-3i | GmGF14j/AtGRF16 |
| GmGF14a/GmGF14u | GmGF14a/Mt14-3-3h | GmGF14d/AtGRF8 |
| GmGF14b/GmGF14k | GmGF14b/Mt14-3-3h | GmGF14d/AtGRF6 |
| GmGF14b/GmGF14u | GmGF14d/Mt14-3-3f | GmGF14h/AtGRF8 |
| GmGF14c/GmGF14e | GmGF14h/Mt14-3-3f | GmGF14h/AtGRF6 |
| GmGF14c/GmGF14r | GmGF14l/Mt14-3-3j | GmGF14c/AtGRF1 |
| GmGF14d/GmGF14 h | GmGF14g/Mt14-3-3j | GmGF14c/AtGRF4 |
| GmGF14f/GmGF14j | GmGF14k/Mt14-3-3g | GmGF14c/AtGRF2 |
| GmGF14g/GmGF14l | GmGF14u/Mt14-3-3g | GmGF14e/AtGRF1 |
| GmGF14g/GmGF14n | GmGF14n/Mt14-3-3c | GmGF14e/AtGRF4 |
| GmGF14g/GmGF14q | GmGF14q/Mt14-3-3c | GmGF14e/AtGRF2 |
| GmGF14k/GmGF14u | GmGF14e/Mt14-3-3e | GmGF14r/AtGRF1 |
| GmGF14l/GmGF14n | GmGF14o/Mt14-3-3b | GmGF14r/AtGRF4 |
| GmGF14l/GmGF14q | GmGF14p/Mt14-3-3b | GmGF14r/AtGRF2 |
| GmGF14m/GmGF14t | | GmGF14g/AtGRF9 |
| GmGF14n/GmGF14q | | GmGF14l/AtGRF9 |
| GmGF14o/GmGF14p | | GmGF14n/AtGRF9 |
| GmGF14r/GmGF14e | | GmGF14q/AtGRF9 |
| | | GmGF14m/AtGRF3 |
| | | GmGF14m/AtGRF7 |
| | | GmGF14m/AtGRF5 |
| | | GmGF14t/AtGRF3 |
| | | GmGF14t/AtGRF7 |
| | | GmGF14t/AtGRF5 |
| | | GmGF14a/AtGRF12 |
| | | GmGF14b/AtGRF12 |
| | | GmGF14k/AtGRF12 |
| | | GmGF14u/AtGRF12 |

predicted to be involved in binding. The number of GmGF14 proteins predicted to cell and binding were very high, suggesting that the *GmGF14* gene family may play a crucial role in protein binding and cell development.

## DISCUSSION

In eukaryotes, the *14-3-3s* were highly conserved and could form homo- or hetero- dimers, which then produced different proteins in a protein complex (*Takahashi et al., 2003*; *Ferl, Manak & Reyes, 2002*). They played important roles in various biological progresses and signal transduction process (*Yoon et al., 2012*; *Wilson, Swatek & Thelen, 2016*). Hence, we carried out genome-wide analysis of *GmGF14* genes by bioinformatics analysis and qRT-PCR to investigate their regulation during development processes and/or stress

**Table 3** List of *Ka*, *Ks* and Ka/Ks values calculated for paralogous *GmGF14* gene pairs.

| Gene 1 | Gene 2 | Ka | Ks | Ka/Ks ratio |
|--------|--------|-----|-----|-------------|
| GmGF14a | GmGF14b | 0.006657464 | 0.098400891 | 0.067656541 |
| GmGF14a | GmGF14k | 0.050954843 | 0.601979458 | 0.084645484 |
| GmGF14a | GmGF14u | 0.052668354 | 0.646064072 | 0.081521874 |
| GmGF14b | GmGF14k | 0.053326335 | 0.624443785 | 0.085398135 |
| GmGF14b | GmGF14u | 0.05474254 | 0.686620201 | 0.07972754 |
| GmGF14c | GmGF14e | 0.041987099 | 0.46411507 | 0.090467003 |
| GmGF14c | GmGF14r | 0.007777127 | 0.133330521 | 0.058329686 |
| GmGF14d | GmGF14 h | 0.014039603 | 0.130901237 | 0.1072534 |
| GmGF14f | GmGF14j | 0.038562958 | 0.163525641 | 0.235822089 |
| GmGF14g | GmGF14l | 0.016897364 | 0.089125506 | 0.189590661 |
| GmGF14g | GmGF14n | 0.095210598 | 1.445569001 | 0.065863752 |
| GmGF14g | GmGF14q | 0.096117116 | 1.432594886 | 0.067093019 |
| GmGF14k | GmGF14u | 0.006698342 | 0.14254911 | 0.046989715 |
| GmGF14l | GmGF14n | 0.08659209 | 1.528599239 | 0.056648 |
| GmGF14l | GmGF14q | 0.087490577 | 1.514174801 | 0.057781028 |
| GmGF14m | GmGF14t | 0.052328317 | 0.201934457 | 0.259135156 |
| GmGF14n | GmGF14q | 0.008325325 | 0.127514764 | 0.065289105 |
| GmGF14o | GmGF14p | 0.015822932 | 0.069501128 | 0.227664399 |
| GmGF14r | GmGF14e | 0.037835791 | 0.428494041 | 0.088299457 |

**Figure 5** Expression analysis of *GmGF14* genes in different tissues. The gene expression values are square-root transformed fragments per kilo-bases per million mapped reads (FPKM). Different colors in map represent FPKM values as shown in bar at top of figure.

responses. Li identified 18 *GmGF14* genes in soybean (*Li & Dhaubhadel, 2011*), however, we found 22 *GmGF14* genes. This may due to the fact that we used a newer version database compared with previous study. Recently, the *14-3-3s* has been reported in several plants, such as Arabidopsis (13), tobacco (17), rice (18), Populus (12), cotton (6), banana (25) and

**Table 4  The number and composition of *cis*-acting regulatory elements of each *GmGF14* gene.**

| Gene | ABRE (ABA) | AuxRR-core (IAA) | TGA-element (IAA) | CGTCA-motif (MeJA) | TGACG-motif (MeJA) | GARE-motif (GA) | P-box (GA) | TATC-box (GA) | TCA-element (SA) | TC-rich repeats (Defense/stress) | LTR (cold) | ARE (anaerobic) | MBS (drought) |
|---|---|---|---|---|---|---|---|---|---|---|---|---|---|
| GmGF14a | | | | | | | | 1 | | | 1 | 1 | |
| GmGF14b | | | | | | | | 1 | | | 1 | | |
| GmGF14c | 1 | | | 2 | 2 | | | | | | | | |
| GmGF14d | 3 | | 1 | 2 | 2 | | | | | 1 | | | |
| GmGF14e | 4 | | | 1 | 1 | | | | | | 1 | 1 | |
| GmGF14f | 2 | 1 | | 1 | 1 | 2 | | | 1 | | | 1 | 1 |
| GmGF14g | 3 | | | 1 | 2 | | 1 | | | | | | |
| GmGF14 h | 7 | | 1 | | | | | | | 1 | | | |
| GmGF14i | 3 | | | 1 | 1 | | 1 | | | | 1 | | |
| GmGF14j | | 1 | 1 | | | 1 | | | | | | 2 | 1 |
| GmGF14k | | | | 1 | 1 | | 1 | 1 | | | | | |
| GmGF14l | 2 | | | 1 | 1 | | 1 | | | | | | |
| GmGF14m | 3 | | | 1 | 1 | | | | | | | | |
| GmGF14n | | | | | | 1 | 2 | | | | | | |
| GmGF14o | | | | | | 1 | 1 | | | | | | |
| GmGF14p | | | | | | 1 | | | | | | | |
| GmGF14q | | | | 1 | 1 | 1 | 1 | | | | | | |
| GmGF14r | 3 | | | 2 | 2 | | | | | | | | 1 |
| GmGF14s | | | | 1 | 1 | | | | | | | | |
| GmGF14t | 1 | | | 2 | 2 | | | | | | | | |
| GmGF14u | | | | 2 | 2 | | 2 | 1 | | | | | |
| GmGF14v | | | | | | | | | | | | | |

# drought

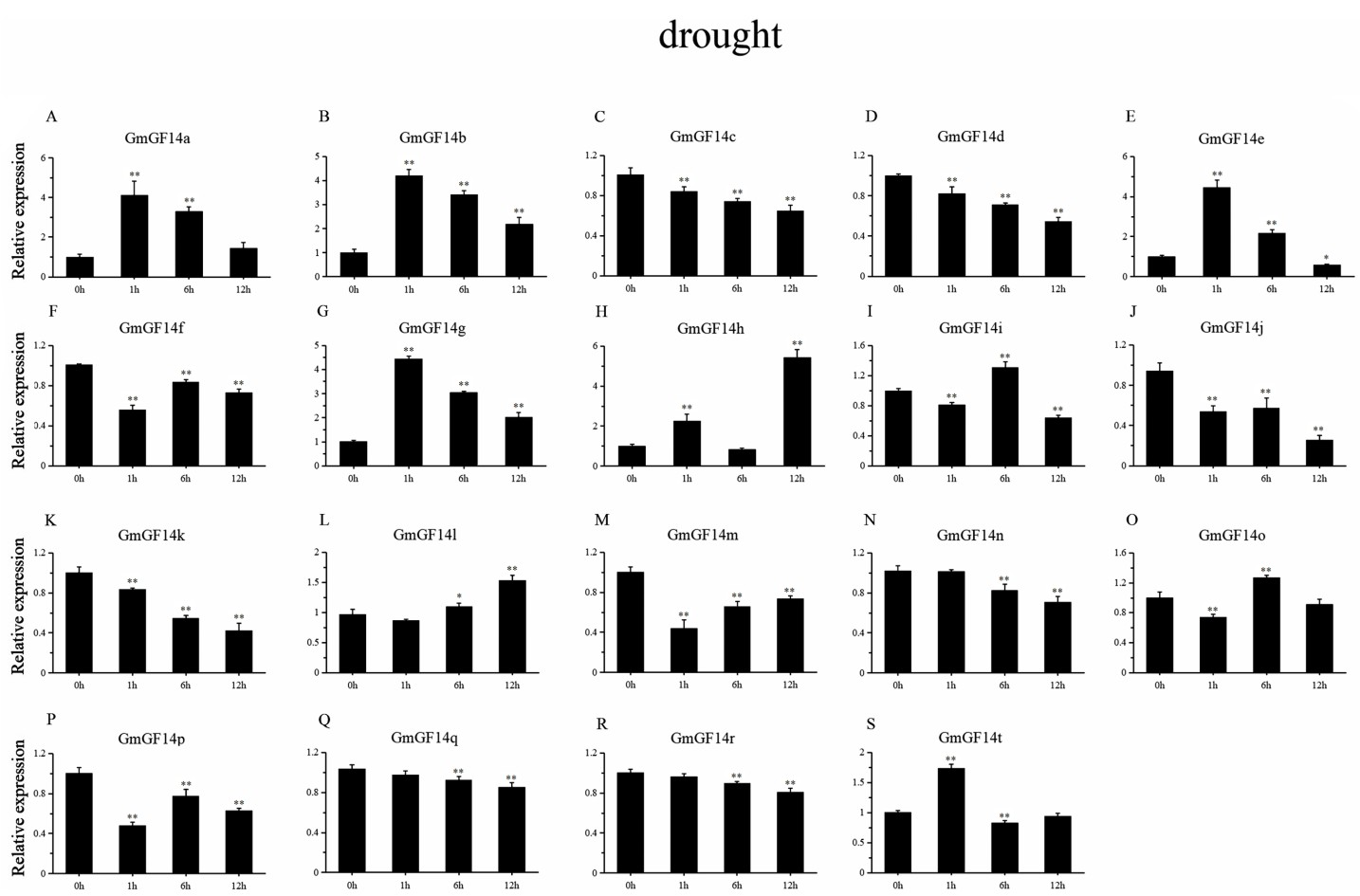

**Figure 6  qRT-PCR analysis reveals *GmGF14* genes under PEG (drought) treatment compared to the controls.** Stress treatments and time course are described in 'Materials and Methods'. (A–S) represent different genes which were used in qRT-PCR analysis. Asterisks on top of the bars indicating statistically significant differences between the stress and counterpart controls ($*p < 0.05$, $**p < 0.01$). Error bars represent SD of biologic replicates.

grape (11) (*Saalbach et al., 1997*; *Ferl, Lu & Bowen, 1994*; *Yashvardhini et al., 2017*; *Tian et al., 2015*; *Zhang et al., 2010*; *Li et al., 2012*; *Cheng et al., 2018*).

In soybean, *14-3-3s* were divided into two groups, $\varepsilon$ group (13 members) and non-$\varepsilon$ group (9 members) based on their phylogenetic analysis. Besides, there is a very close relationship between soybean and *M. truncatula*, suggesting that the *14-3-3* family members in legumes are relatively conserved. In addition, $\varepsilon$ group *GmGF14* genes have much more exons/intron than non-$\varepsilon$ group genes, while the first intron of non-$\varepsilon$ group was longer than that of $\varepsilon$ group. Besides, the members in $\varepsilon$ group contained eight motifs, while non-$\varepsilon$ group members had less, usually 3-4 motifs. Furthermore, protein structure analysis show that compared with other species such as banana, grape and rice, the members of *14-3-3s* had ten typical antiparallel $\alpha$-helices (*Yashvardhini et al., 2017*; *Cheng et al., 2018*; *Li et al., 2012*). The result of *14-3-3* proteins was different from that in other species, that might

salt

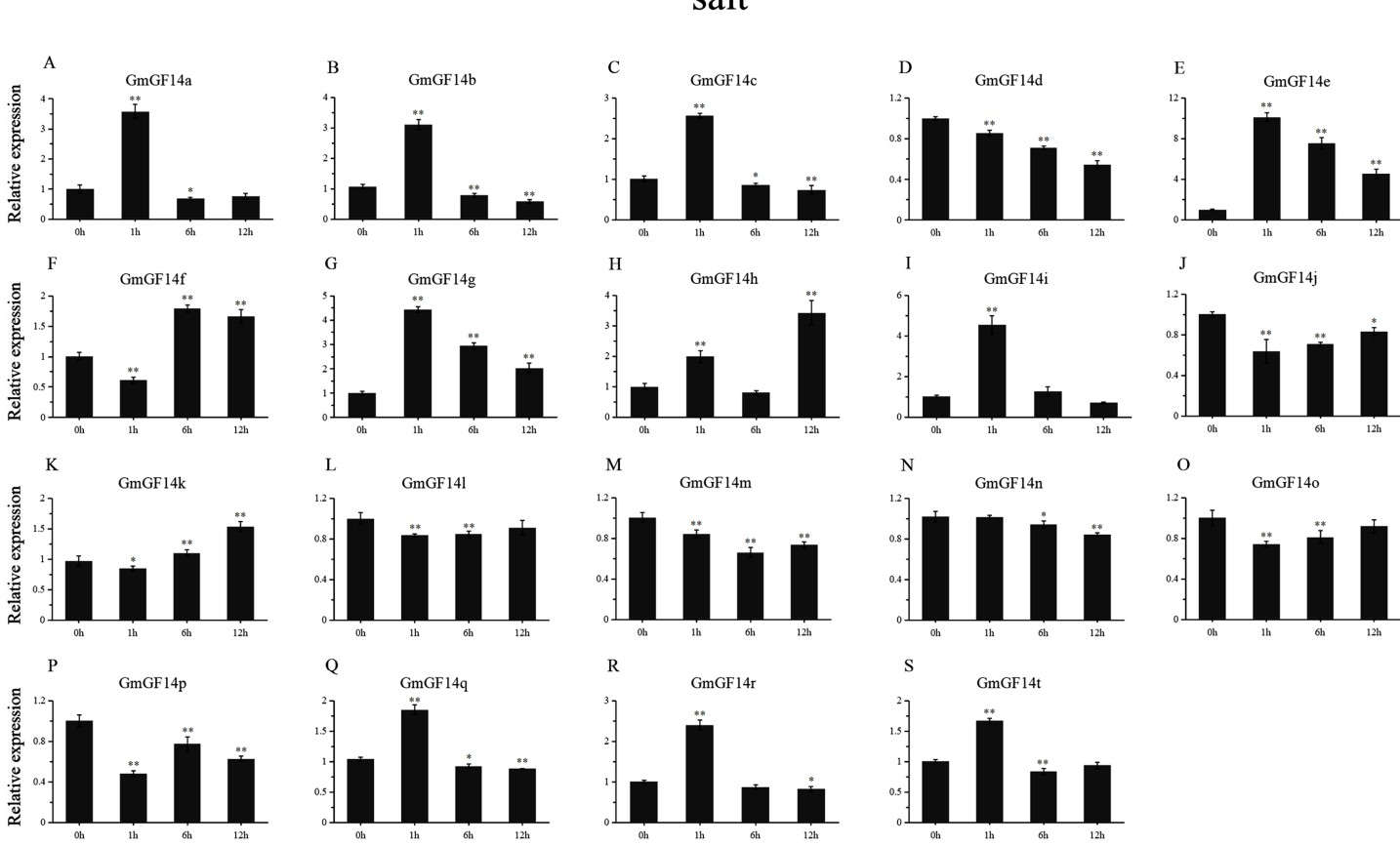

**Figure 7** **qRT-PCR analysis reveals.** Stress treatments and time course are described in 'Materials and Methods'. (A–S) represent different genes which were used in qRT-PCR analysis. Asterisks on top of the bars indicating statistically significant differences between the stress and counterpart controls (*$p < 0.05$, **$p < 0.01$). Error bars represent SD of biologic replicates.

due to the soybean genome has undergone two gene duplication events and has more gene diversity in the process of evolution (*Wang et al., 2017*).

Gene duplication events is important in gene family expansion and could gain functional diversity during evolution, including tandem, transposition and segment duplication events (*Kaessmann, 2010*). There were 19 gene pairs involved in segment duplication, while no tandem duplication event occurred in *GmGF14s*, indicating that the segment duplication maybe the major gene duplication for this gene family's expansion (*Cheng et al., 2018*). Among them, ε group had more gene duplication events (14/19; 73.68%) than non-ε group (5/19; 26.32%) in soybean. In addition, we calculated the *Ks* value of each paralogous pairs, and found the most recent duplication event in soybean appeared between 5 to 20 MYA, which is consisting with the recent whole genome duplication (WGD) event in soybean (*Wang et al., 2017*). The *Ka/Ks* of all the *GmGF14* gene pairs were less than 0.3, suggesting that they were evolved in mainly under strong purifying selection. This result was similar to other plants, meaning that *14-3-3* genes evolved more slowly at the protein level in plants, and they have a conserved evolutionary pattern in *GmGF14* genes.

cold

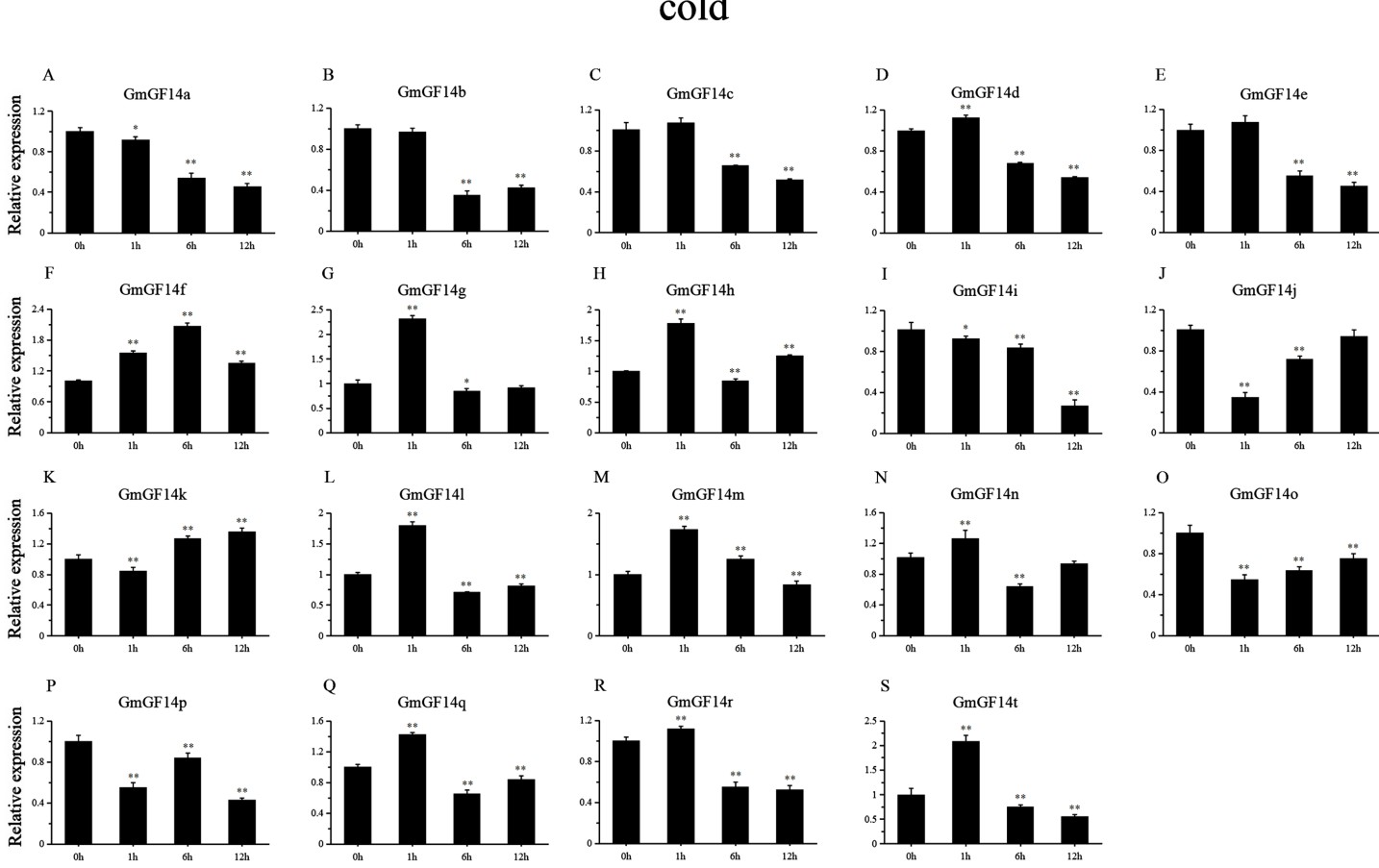

**Figure 8** **qRT-PCR analysis reveals.** *GmGF14* genes under cold treatment compared to the controls. tress treatments and time course are described in 'Materials and Methods'.(A–S) represent different genes which were used in qRT-PCR analysis. Asterisks on top of the bars indicating statistically significant differences between the stress and counterpart controls (*$p < 0.05$, **$p < 0.01$). Error bars represent SD of biologic replicates.

It has been reported that the *14-3-3* genes were expressed in different tissues in many plants. *PvGRFr* might involve in flower development based on the expression patterns in switchgrass (*Wu et al., 2016*). In banana, the expression quantity of most *MaGRFs* were accumulated during fruit ripening obviously (*Li et al., 2012*). The expression levels of most *GmGF14* genes in vegetative organs were higher than that of reproductive organs in plants, suggesting that *14-3-3* genes may participate in morphogenesis directly or indirectly. In soybean, *14-3-3* genes involved in nodule mature, they can affect the formation of the early nodule development when the expression levels of *SGF14c* and *SGF14l* reduced (*Radwan et al., 2012*). In *M. truncatula*, *Mt14-3-3* genes were involved in rhizobium infections, *Mt14-3-3c* was involved in the early stage of nodule formation (*Chen et al., 2006*). These results suggesting that the *14-3-3* gene family has various functions, they were similar to the results of the GO enrichment. For the GO enrichment, *GmGF14* genes were mainly concentrated in cell development and protein binding. Except this, different *GmGF14* genes had similar expression patterns in different tissues. For example, paralogous pairs

*GmGF14a/b*, *GmGF14k/u*, and *GmGF14o/p* had similar expression patterns in most tested tissues, meanwhile, they also had gene duplication relationship, indicating that they might have similar functions in different tissues, in accordance with the results of GO enrichment.

More and more evidences had suggested that *14-3-3* genes response to environmental stimuli in many plants (*Xu & Shi, 2006*; *Chen et al., 2006*; *Li et al., 2015*). Plant *14-3-3* genes are signal moderators, playing an important role in response to abiotic stress (*Li et al., 2015*). Overexpression of *AtGRF9* resulted in more carbon distribution from the shoot to the root, and enhanced the drought tolerance of plant by increasing proton secretion in the root growing zone (*He et al., 2015*). As homologous gene of *AtGRF9*, *GmGF14g* was up regulated during the treatment, and the 1 h drought stress treatment caused a threefold increase of its expression level (its expression increased 3-fold with 1 h drought stress treatment). In tomato, transcription level of four *14-3-3* genes were significantly up-regulated under salt stress (*Xu & Shi, 2006*). In this study, nine genes (*GmGF14a/b/c/e/g/i/q/r/t*) were first up-regulated and then decreased after salt treatment, indicating that soybean *14-3-3* genes have different regulatory mechanisms under stress. In addition, many *14-3-3* genes (e.g., *GmGF14b/c/g/j*) of soybean changed distinctly under cold treatment, most of the gene expression levels decreased at 6 h and 12 h treatment time points, suggested that they might play a potential role in responding to cold stress. The results of GO enrichment suggested that *GmGF14* genes can respond to stimuli.

The ABA signaling pathway is a major pathway in response to the drought, salt and cold stresses (*Zhang et al., 2006*; *Yu & Qi, 2017*). *14-3-3s* promoter region contain ABRE promoters, and they could response to stresses directly or indirectly by involving in the ABA signal pathway. In addition, four *cis*-elements of *GmGF14* genes (TC-rich repeats, ARE, MBS and LTR) involved in responding to different abiotic stresses, while other eight *cis*-elements were involved in multiple plant hormone stress responses. Taken together, these results reported that *GmGF14s* may have various functions, including regulate plant growth and response to abiotic stresses.

## CONCLUSIONS

All 22 *GmGF14s* were classified into the $\varepsilon$ group and non-$\varepsilon$ group based on their phylogenetic relationship among *A. thaliana*, *O. sativa* and *M. truncatula*. Gene structure and duplication event showed that *14-3-3* gene family was relatively conserved. RNA-seq and qRT-PCR were used to explore the function of *GmGF14s*. The expression levels of most *GmGF14s* showed that they could response to multiple stresses. In summary, these results suggest the potential roles of *GmGF14* genes in plant development and multiple stress responses, therefore provide scientific references for the further study of *GmGF14* genes' function.

## ACKNOWLEDGEMENTS

The authors would like to thank the Key Laboratory of Crop and Livestock Molecular Breeding of Heilongjiang Province for providing plenty of helpful manpower and material support.

### Funding

This work was supported by the National Key R&D Program for Crop Breeding (2016YFD0102105), the Science and Technology Program for Innovation Talents of Harbin (2014RFQYJ016, 2014RFXYJ011) and the Project of the Heilongjiang Academy of Agricultural Sciences (HAAS) (2017BZ12). The funders had no role in study design, data collection and analysis, decision to publish, or preparation of the manuscript.

### Grant Disclosures

The following grant information was disclosed by the authors:
National Key R&D Program for Crop Breeding: 2016YFD0102105.
Science and Technology Program for Innovation Talents of Harbin: 2014RFQYJ016, 2014RFXYJ011.
Project of the Heilongjiang Academy of Agricultural Sciences (HAAS): 2017BZ12.

### Competing Interests

The authors declare there are no competing interests.

### Author Contributions

- Yongbin Wang performed the experiments, analyzed the data, contributed reagents/materials/analysis tools, prepared figures and/or tables, authored or reviewed drafts of the paper, approved the final draft.
- Lei Ling analyzed the data, prepared figures and/or tables, authored or reviewed drafts of the paper, approved the final draft.
- Zhenfeng Jiang performed the experiments, analyzed the data, contributed reagents/materials/analysis tools, prepared figures and/or tables, approved the final draft.
- Weiwei Tan Zhaojun Liu performed the experiments, prepared figures and/or tables, approved the final draft.
- Licheng Wu, Shanyong Xia and Jun Ma performed the experiments, contributed reagents/materials/analysis tools, prepared figures and/or tables, approved the final draft.
- Yuanling Zhao performed the experiments, prepared figures and/or tables, approved the final draft.
- Guangjin Wang conceived and designed the experiments, prepared figures and/or tables, approved the final draft.
- Wenbin Li conceived and designed the experiments, authored or reviewed drafts of the paper, approved the final draft.

### Data Availability

The raw measurements are available in the Supplementary Files.

## Supplemental Information

Supplemental information for this article can be found online at http://dx.doi.org/10.7717/peerj.7950#supplemental-information.

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
