# Peer review of "Genome-wide identification and expression analysis of the 14-3-3 gene family in soybean (Glycine max)"

_PeerJ, doi:10.7717/peerj.7950_

## Round 0.1 · original submission · Major Revisions

Dear authors,

Your manuscript has been assessed by two reviewers and myself as academic Editor. Based on the comments received, I think the findings of the study are interesting and the manuscript can be reconsidered for publication only after major revisions are incorporated. Both reviewers have found significant flaws in the paper. In particular, insufficient literature references and method details are provided, and the data presented is not statistically sound. In addition, you are asked to improve the manuscript by removing spelling mistakes and grammatical errors throughout the paper. When preparing your revised manuscript, I suggest you to carefully address reviewer comments and submit a detailed response to each comment. Please address all concerns of the reviewers and submit a revised version of the manuscript.



# ·

Basic reporting

This manuscript provides a characterization of the Soybean 14-3-3 gene family in soybean (Glycine max).
As already an attempt was taken in this direction, where, Li X, Dhaubhadel S (Li X, Dhaubhadel S. Soybean 14-3-3 gene family: identification and molecular characterization. Planta. 2011;233:569–82. pmid:21120521) identified eighteen 14-3-3 genes in soybean. So further, in the current study authors have reported 22, 14-3-3 genes in soybean, and also performed phylogeny and expression analysis of these genes.

But i have some major concerns. Below are the issues that need to be addressed by the authors before the manuscript can be accepted for publication.

Poor English with many grammatical and spelling mistakes throughout the manuscript. Which need to be improved.

More details about the results and their statistical significance is required in the Figure legends.

Literature references
sufficient field background/context are not provided,
Authors didn't mention about the important work already published about "Soybean 14-3-3 gene family: identification and molecular characterization"
by
Li X, Dhaubhadel S. Soybean 14-3-3 gene family: identification and molecular characterization. Planta. 2011;233:569–82. pmid:21120521

Experimental design

Original primary research within Aims and Scope of the journal.

Research question well defined, relevant & meaningful. It is stated how research fills an identified knowledge gap.

Rigorous investigation was performed to a high technical & ethical standard except in few cases.

Methods are not described with sufficient details & information to replicate
Like in RNA extraction and Quantitative real-time PCR (qRT-PCR)
No details has been provided about the biological amples and PCR conditions

the Authors need to specify about the biological samples taken in the relative expression analysis. Also in the material method and results sections i can not see any details about statistical significance of the results.

Validity of the findings

Data is not statistically sound

Reviewer 2 ·

Basic reporting

The English language should be improved throughout the manuscript to allow the reader to understand the text. The manuscript is difficult to read and there are many expressions that need to be corrected. For example; Lines 300-301.

The Introduction, background and discussion are in general well referenced, although some literature related to 14-3-3 in soybean is missing. For example; Li and Dhaubhadel (2011) and (2012).

In general Figures are relevant and have good quality. I suggest black labels in Figures related to qRT-PCR analysis to ensure proper visualization. In addition, in Figure 5 the different colors represent transcript abundance. Are they FPKM values? This should be explained in more detail in the Figure Legend and in methods.

Experimental design

The Methods are in general well described. However some more details are needed for the measurement of transcript abundance and qPCR analysis should at least have 3 biological replicates and statistical analysis should be included to determine the significance of the data.

Validity of the findings

The results obtained are interesting; however more information can be obtained to strengthen the findings. In addition, the RT-qPCR should be repeated with at least 3 biological replicates.

Previous papers in soybean describe the presence of 18 14-3-3 genes. The authors should indicate this difference in the text and explain why they found 22.

The number of genes pairs involved in segmental duplication should be indicated in the result section.

The authors could include a Figure or table showing the segmental duplication pairs, the % of identity, and their expression patterns in different tissues/organs and abiotic stress. The fact that similar expression patterns are present in the duplicated pairs is interesting and could be shown more clearly.

The expression patterns of some members of segmental pairs during abiotic stress are missing. What were the criteria to use 19 and not 22 genes?

The expression pattern of the 14-3-3 from M. truncatula could also be included since expression data is available and the results could contribute to the discussion.

In addition to ABA responsiveness, some other cis-regulatory elements are also overrepresented in the 14-3-3 gene members. This could be discussed in more detail.

Minor comments
Please change “gene replication” for “gene duplication”

Additional comments

In this manuscript, the authors analyzed the 14-3-3 gene family in the soybean genome, compared their phylogenetic relationship with 14-3-3 from Arabidopsis, rice and Medicago truncatula, and classified them according to conservative motifs and gene structures. They show the occurrence of segmental duplication events and the evolution of the soybean 14-3-3 gene family comparing them with Arabidopsis and M. truncatula. Using the publicly available RNA-Seq data, the authors analyzed the expression patterns of this gene family in different tissues and organs. The authors also analyzed the expression pattern in response to abiotic stress, including drought, salt and cold stress. The results show that 14-3-3 soybean genes are higher expressed in vegetative organs compared to reproductive organs and that they are differentially regulated during abiotic stress. This manuscript would be of general interest. However, there are major comments that need to be addressed.

---

## Round 0.2 · Minor Revisions

Dear Yongbin Wang,

Thank you for providing the revised manuscript. The previous reviewers were unavailable, so the revised manuscript has been assessed by myself as an Academic Editor. I appreciate that the authors have carefully addressed reviewer comments and have submitted a detailed response to each comment. The revised version has certainly been improved. In particular, suggested literature references are added, missing method details are provided, and the data presented here is now statistically sound. However, there are a few minor changes that are essential to implement before publication:

1. In general, I found legends of the tables/figures very less informative/incomplete. For e.g. legend of Fig. 4 is "Chromosome location and duplication events analysis in Glycine max". In addition to this sentence in the legend, more details are required to fully understand the figure. For e.g. details of scale bar to represent chromosome, what do blue and red lines correspond to, how many duplication events in total did author found etc. I insist authors to please review and elaborate legends of tables/figures, both in the main paper and supplementary to provide as much as possible details.
2. Though spelling mistakes and grammatical errors are improved in the revised version, however, I still can spot typos/grammatical errors at few places. For e.g. ExPASy is written as “ExPasy” throughout the manuscript, and abiotic is written as “abotic” in legend “Table S3 Similarity of duplication gene pairs under abotic stresses”, “V” should not be in the upper case to represent a version in Phytozome V12.1 database etc. I insist authors to carefully go through the entire manuscript to remove such typos and additional grammatical errors.

Finally, I encourage authors to submit a revised version of the manuscript to be considered for final publication in PeerJ after incorporating the above suggestions.

Best regards,
Shalu Jhanwar

---

## Round 0.3 · Minor Revisions

Dear Dr. Wang,

Thank you for providing the revised manuscript. The revised manuscript has been assessed by myself as an academic editor and Gerard Lazo, the Section Editor. The revised version has certainly been improved. However, there are a few changes that are essential to implement before publication. Below are the specific points from Dr Lazo:

a) Considering that the authors are trying to characterize well-documented gene family and to add value to their assessments it is requested that the assignment of gene ontology (GO) terms be applied to the data to differentiate the members based on molecular function, biological process, and cellular component terms. The differential expression data was limited to a heat map with no real assessment of the value of the observed data. The changes in gene expression levels were related in general terms with no focus to known traits or phenotypes associated with soybean to be connected to the data. Outlining such descriptions require a tractable annotation rather than casual verbal descriptions, and GO annotations would be valuable for this purpose.

b) Assignment listings for the GO terms might be easily added to the defined sequence table. Journal manuscripts are often scanned by text-mining software that locates and extracts core data elements, like gene function. Adding standard ontology terms, such as the Gene Ontology (GO, geneontology.org) or others from the OBO foundry (obofoundry.org) can enhance the recognition of your contribution and description.

c) It would be of value if the extracted sequences from the Phytozome might also be created as a supplemental FASTA file to assist readers in validation for observations. The proposed exon/intron structures would be difficult to re-create based on the structures provided in Figure 3.

d) There were several areas where the language was a bit rough due to some language barriers. It would be worthwhile having a professional service review the manuscript for improving the readability. There were a number of areas that required improvement, and I have mentioned a few; some reflection to providing an improved version is required.

Finally, I encourage authors to carefully address the revisions and submit a revised version of the manuscript to be considered for final publication in PeerJ.

Best regards,
Shalu

---

## Round 0.4 · Minor Revisions

Dear Dr. Wang,

Thank you for providing the revised manuscript. The revised manuscript has been assessed by myself as an academic editor. I appreciate that the authors have carefully addressed the previous revisions. The revised version has certainly been improved. However, below are the specific areas that required improvement before publication:

a) I suggest authors strengthen the abstract by rephrasing informal and repetitive sentences. For e.g.
a. line 30 – “Previous studies identified 18 GmGF14 genes in soybean, which are different from ours”.
b. Be specific and comprehensive with the type/name of stress responses in sentence at line 35-36 – “Moreover, the expression level of most GmGF14s changed obviously in multiple stress responses, suggesting that they have the abilities of responding to multiple stresses”.
b) Line 34 - no space between full stop and morphogenesis in “regulator of soybean morphogenesis .”. Line 330 - no space between comma and stresses in “abiotic stresses ,”.
c) Typo in line 70 – “attentions focuse on soybean GmGF14s”. Here focuse should be “focus”.
d) Line 319 – “3 fold” should be “3-fold”
e) Line 99 – reference missing for Jalview. Also, please mention the name of the method used to perform multiple alignments of proteins using Jalview. For e.g. MAAFT, ClustalW etc.
f) Line 153-154 - incomplete sentence as “The GO annotation of the GmGF14 genes by using WEGO 2.0 website (http://wego.genomics.org.cn/)”.
g) Line 201 -202 - typo in the sentence “Two or more GmGF14 genes matched to one AtGRF gene or Mt14-3-3 gene, impling that these genes might play key roles in the GmGF14 genes’ expansion during evolution”. “impling” should be replaced as “implying”.
h) Line 242 – typo in “significantly changed”. It should be “significantly changed”.
i) The result section of “Gene Ontology Enrichment” is very poorly explained. Please elaborate in relation to the present findings.
j) Figure S2 is not very informative as it contains a very broad term as “protein binding” without any specific terms. If there’s only “protein binding” term, then it can easily be included directly in the form of text. Also, the scale bar is ambiguous with no information if the terms are significantly enriched or not? Legend for fig. S2 is poorly written without providing sufficient information. I would suggest authors to update the fig. and legend with specific details.
k) Both the sentences in the method section “Gene Ontology Enrichment” are incomplete. Moreover, the details of statistics/thresholds used to obtain significant enrichment are completely missing. This data is required to assess the importance of the results. Please rephrase the sentences and add this important information.
l) I would insist authors to discuss the results of Table S9 (GO terms of the GmGF14 genes) in the light of the present study. That way it will be extremely useful for readers to understand the importance of GO analysis in studying soybean GmGF14s.

Please be aware that the line numbers correspond to the .docx version of the revised manuscript. Finally, I encourage authors to carefully address the revisions and submit a revised version of the manuscript to be considered for final publication in PeerJ.

Best regards,
Shalu

---

## Round 0.5 · accepted · Accept

Dear Dr. Wang,

I can read that you have addressed the section editor concerns, and you have added the requested information on GO annotations. This may be of value for those that attempt to re-visit this research question. Please consider this manuscript accepted.

Best regards,
Dr. Shalu Jhanwar